# Molecular Characterization of Non-responders to Chemotherapy in Serous Ovarian Cancer

**DOI:** 10.3390/ijms20051175

**Published:** 2019-03-07

**Authors:** Megan E. McDonald, Erin A. Salinas, Eric J. Devor, Andreea M. Newtson, Kristina W. Thiel, Michael J. Goodheart, David P. Bender, Brian J. Smith, Kimberly K. Leslie, Jesus Gonzalez-Bosquet

**Affiliations:** 1Division of Gynecologic Oncology, Department of Obstetrics and Gynecologic, University of Iowa Hospitals and Clinics, Iowa City, IA 52242, USA; megan-e-mcdonald@uiowa.edu (M.E.M.); andreea-newtson@uiowa.edu (A.M.N.); michael-goodheart@uiowa.edu (M.J.G.); david-bender@uiowa.edu (D.P.B.); 2Compass Oncology, Portland, OR 97227, USA; Erin.Salinas@compassoncology.com; 3Department of Obstetrics and Gynecology, University of Iowa Hospitals and Clinics, Iowa City, IA 52242, USA; eric-devor@uiowa.edu (E.J.D.); kristina-thiel@uiowa.edu (K.W.T.); kimberly-leslie@uiowa.edu (K.K.L.); 4Holden Comprehensive Cancer Center, University of Iowa Hospitals and Clinics, Iowa City, IA 52242, USA; brian-j-smith@uiowa.edu; 5Department of biostatistics, University of Iowa College of Public Health, Iowa City, IA 52242, USA

**Keywords:** serous ovarian cancer, chemotherapy response, TCGA, *iCLusterPlus*, Unsupervised clustering

## Abstract

Nearly one-third of patients with high-grade serous ovarian cancer (HGSC) do not respond to initial treatment with platinum-based therapy. Genomic and clinical characterization of these patients may lead to potential alternative therapies. Here, the objective is to classify non-responders into subsets using clinical and molecular features. Using patients from The Cancer Genome Atlas (TCGA) dataset with platinum-resistant or platinum-refractory HGSC, we performed a genome-wide unsupervised cluster analysis that integrated clinical data, gene copy number variations, gene somatic mutations, and DNA promoter methylation. Pathway enrichment analysis was performed for each cluster to identify the targetable processes. Following the unsupervised cluster analysis, three distinct clusters of non-responders emerged. Cluster 1 had overrepresentation of the stage IV disease and suboptimal debulking, under-expression of miRNAs and mRNAs, hypomethylated DNA, “loss of function” *TP53* mutations, and the overexpression of genes in the *PDGFR* pathway. Cluster 2 had low miRNA expression, generalized hypermethylation, *MUC17* mutations, and significant activation of the HIF-1 signaling pathway. Cluster 3 had more optimally cytoreduced stage III patients, overexpression of miRNAs, mixed methylation patterns, and “gain of function” *TP53* mutations. However, the survival for all clusters was similar. Integration of genomic and clinical data from patients that do not respond to chemotherapy has identified different subgroups or clusters. Pathway analysis further identified the potential alternative therapeutic targets for each cluster.

## 1. Introduction

Epithelial ovarian cancer is the fifth leading cause of cancer death among women in the United States and it has the highest mortality rate of all gynecologic cancers [1]. The majority of patients present with advanced disease at diagnosis. While most of the patients respond to the initial combination treatment of surgical debulking and platinum-based chemotherapy, nearly one-third of patients will not respond. Significant effort has been expended to define platinum resistance in epithelial ovarian cancers at both the histologic and biologic levels. For instance, increased chemoresistance has been described in low grade tumors [2] and in mucinous and clear cell histologic subtypes [3,4]. Epithelial ovarian cancer patients with germline and/or somatic *BRCA* mutations have improved survival, which is likely due to increased sensitivity to platinum-based DNA damming chemotherapy [5]. Various mechanisms for platinum resistance have been described, both spontaneous and acquired [6,7,8,9,10], though the exact mechanism for resistance is potentially tumor-dependent. 

Despite this improved understanding of platinum resistance on the molecular level, clinical outcomes remain poor for ovarian cancer patients. Recently published data from The Cancer Genome Atlas (TCGA) high-grade serous ovarian cancer (HGSC) dataset showed that responders to initial chemotherapy experience a more than two year increase in median overall survival when compared to non-responders (*p* < 10^−14^) [11]. Previous studies have used epithelial ovarian cancer patients’ serum biomarker data to predict the response to initial chemotherapy with an area under the curve (AUC) of 70–77%. When combined with clinical data, the classification capacity in these studies could increase the AUC to 91% [12,13]. However, these prediction models are limited in their clinical application due to the heterogeneity of histologic subtype and stage, as well as the lack of validation in independent datasets. In a previously published prediction model using the TCGA dataset, we identified a 34-gene signature that predicts chemosensitivity specifically in HGSC, with an AUC approaching 80%. This 34-gene signature was then validated in six independent gene expression datasets [14]. 

However, as our ability to predict chemo-response becomes more accurate, the lack of alternative or adjuvant therapies for patients who are predicted to fail standard first-line platinum-based therapies has become acutely apparent. Through the integration of clinical and molecular data, our objective in this pilot study was to characterize the HGSC patients that do not respond to initial chemotherapy, which would, in turn, inform the design of personalized treatment combinations. 

## 2. Results

### 2.1. Data Preprocessing 

Evaluation of clinical data revealed that non-responders were more often diagnosed at a later stage (*p* = 0.01) and had suboptimal surgical outcomes (*p* < 0.001), but there were no differences in histological grade or usage of platinum-based chemotherapy when compared with responders. In Table 1 we described clinical characteristics of all non-responders. Patients were categorized according to the clusters that they belonged following the analysis with the integrative cluster method, *iClusterPlus*. None of the clinical characteristics were statistically significant between the three groups, and they were therefore not accounted for in the clustering process. 

Results of the univariate analyses between responder and non-responder groups for all types of biological data are represented in Figure 1. All of the significant variables between groups were used for clustering analysis: 1023 differentially expressed genes, 960 differentially methylated promoters, 21 differentially expressed miRNAs, 56 somatic mutations, and more than 8000 gene copy number alterations (CNA). Initial evaluation with the clustering tool demonstrated that the miRNA variables did not influence the clustering process; thus, only the other four classes of biological data were included in the final analysis: gene expression, DNA methylation, somatic mutations, and gene copy number alteration.

### 2.2. Integrative iClusterPlus Analysis 

Eighty-eight non-responders from TCGA who had complete information for outcome, gene expression, mutation analysis, gene copy number, and DNA methylation were included in the cluster analysis. The optimization of the clustering method identified three differentiated clusters within non-responders (Figure 2). To build the final model, we used a threshold, or cut-off value, which selected the most discriminative features for the three-cluster model solution. The threshold was the 95th percentile. Only those features that passed this threshold were included in the representation of the final three-cluster model. Clinical information, including surgical outcomes along with type of *TP53* somatic mutation, and independently significant miRNAs were added to each cluster in a supervised manner for further characterization (Figure 3). 

A significant portion of Cluster 1 patients had > 2cm of residual visible disease following their primary debulking surgery. Additionally, they more often received suboptimal initial treatment. When classifying the *TP53* somatic mutations within each cluster, the most loss-of-function *TP53* mutations were seen in Cluster 1. Low levels of miRNA expression, significant hypomethylation, a high rate of somatic mutations, and CNA at chromosome 19 further characterized cluster 1. 

Cluster 2 patients have mixed clinical characteristics and types of *TP53* mutations. However, this group has extremely low levels of miRNA expression and higher expression of representative genes, as well as greater DNA hypermethylation, when compared to Cluster 1 and 3 patients. Further, unlike Clusters 1 and 3, there were no somatic mutations in the *DNAH5* and *ODZ1*. Cluster 2 patients were more often diagnosed at stage III, received optimal primary treatment, and had optimal cytoreductive surgery, with the greatest percentage of completely cytoreduced patients as compared to other clusters. Cluster 3 patients had the greatest number of “oncogenic” gain-of-function (GOP) *TP53* mutations. Cluster 3 was further characterized by high miRNA expression and DNA hypermethylation.

### 2.3. Pathway Enrichment Analysis 

Pathway analysis findings were significant for the overrepresentation of the platelet derived growth factor receptor (*PDGFR*) in different pathways in Cluster 1 (Table 2). Overrepresentation of the HIF-1 signaling pathway was seen in Cluster 2; the Wnt/β-catenin pathway was close but not statistically significant in this cluster (*p*-value = 0.089). Pathways that are involved in cellular senescence were found to be overrepresented in Cluster 3. Despite significant variation in clinical and molecular profiles between the clusters, survival differences were not observed among clusters (Figure 4). 

## 3. Discussion

While understanding the biologic mechanism for platinum resistance in ovarian cancer is extremely valuable, this information has not yet been translated into effective therapeutic strategies. To date, the most widely used molecular inhibitors in the treatment of ovarian cancer are the *VEGF* inhibitor bevacizumab and poly (ADP-ribose) polymerase (PARP) inhibitors. Even with the incorporation of these drugs into the accepted treatment paradigm for ovarian cancers, molecular analysis of tumors is not routine. Here, our goal was not to assess the mechanism of platinum resistance but, rather, to characterize non-responders. Our unsupervised cluster analysis of HGSC patients that did not respond to primary platinum-based chemotherapy revealed three distinct groups. 

Even though specific clusters did not predict survival, unique differences between the clusters could eventually be used for treatment planning. For example, non-responders in Cluster 1 were more often diagnosed at stage IV and receive suboptimal treatment. From the progression-free survival (PFS) and overall survival (OS) results of ICON7 and GOG 218, we now know that patients with poor prognosis disease, defined in the studies as those with stage IV disease, inoperable stage III disease, or suboptimally debulked stage III disease, have a slightly higher PFS and OS advantage with the addition of bevacizumab to standard chemotherapy [16,17]. 

We hypothesized that the combination of cluster analysis, clinical and mutational characteristics, and pathway analyses may reveal some of the biology behind each cluster. Indeed, pathway analysis of molecular characteristics can be used to gain insight into the common biological processes that contribute to disease pathophysiology for each cluster [18]. For example, Cluster 1 seems to overrepresent pathways that are involved in growth factor signaling (*PDGFR* and *VEGF*), which contributes to cell proliferation. Pathways that regulate cellular survival, particularly in the setting of stress (HIF-1 signaling pathway and cellular senescence), dominate Cluster 2. Finally, the significantly altered pathway in Cluster 3 was related to cellular senescence. Not only do these distinct alterations hint at the potential mechanisms of chemoresistance, but they can also be used to suggest alternative treatments to override the dominant signaling pathways. 

Since Cluster 1 tumors have an overrepresentation of growth factor signaling pathways, a logical treatment choice for these patients would be a tyrosine kinase or angiokinase inhibitor. Indeed, a recent phase 3 trial using nintedanib (an oral triple angiokinase inhibitor of VEGF receptor, PDGFR, and fibroblast growth factor receptor), in addition to platinum based chemotherapy showed improved PFS for ovarian cancer (17.2 months vs 16.6 months; *p* = 0.024) [19]. Additionally, a recent phase II trial for platinum-resistant ovarian cancer patients has identified a group of patients with increased PFS after treatment with nintedanib [20]. In ICON6, the use of cediranib, another oral angiokinase inhibitor that targets VEGFR and PDGFR, in addition to platinum-based chemotherapy for platinum sensitive, recurrent ovarian cancer showed a PFS benefit of almost three months (*p* < 0.0001) [21]. Through the identification of Cluster 1 patients prior to initial treatment, we could potentially single out those patients that are likely to benefit from the addition of angiokinase inhibitors or bevacizumab to chemotherapy in the front-line setting and potentially improve survival.

The overrepresented pathways in Cluster 2 are involved in cell survival. Notably, the HIF-1 signaling pathway is intimately related to the PI3K/AKT/mTOR pathway, which is one of the most important signaling pathways in cell survival [22]. In pre-clinical studies, the chemoresistant ovarian cancer cells are highly sensitive to mTOR inhibitors [23]. However, Phase II trials in ovarian cancer, including resistant cases, have demonstrated a diverse range of responses to mTOR inhibitors, such as temsirolimus [24], indicating that better criteria are necessary to determine which patients are the best candidates for mTOR inhibitors. Our study may shed insight into this important clinical question. In the pathway enrichment analysis of Cluster 2, the Wnt/β-catenin signaling pathway was not significant (*p*-value = 0.09), but cellular senescence was (*p*-value = 0.001). It has been reported that Wnt signaling antagonizes oncogene-induced cellular senescence as a tumor suppression mechanism *in vivo* [25]. Suppressing Wnt signaling pathway may improve in response to treatment in some tumors. Not only are multiple Wnt inhibitors under preclinical development, but Wnt inhibitors, such as LGK974 and OMP18R5, have entered phase 1 clinical trials for solid tumors with aberrant Wnt signaling [26], and PRI-724 is currently being used in a phase II trial in metastatic colon cancer [27]. Additionally, recent phase 1 evaluation of the Wnt/β-catenin signaling pathway within melanoma has shown that overactivation of the Wnt pathway conveys resistance to immunotherapy with anti-PD-L1 monoclonal antibody [28], a treatment that has been recently extrapolated for use in gynecologic tumors [29,30]. 

The most significantly overrepresented pathway in Cluster 3 was the cellular senescence pathway. In cancer transformation, the cell needs to bypass cellular senescence to become immortal [31]. *TP53*, which is a key tumor suppressor gene, is one of the most well-known components of this pathway. Gain-of-function *TP53* mutations were most frequent in Cluster 3 patients. In addition to targeted agents that are focused on the reactivation of null *TP53*, there are many investigational compounds that induce gain-of-function (GOF) mutant *TP53* degradation, such as histone deacetylase (HDAC) inhibitors [32]. As described by Yan et al., the activity of HDAC inhibitors extended to decreasing not only GOF *TP53*, but also wild-type *TP53* expression, indicating that knowledge of the mutant type of *TP53* is critically important before the application of these targeted therapies [33]. 

This study is circumscribed by available TCGA data and the potential systematic biases from retrospective data involving selected patients. Therefore, before we can apply this classification model to patients with advanced stage HGSC and design new alternative treatments, it is necessary to validate the classification in a prospective study and within other, representative datasets. We are currently recruiting patients and processing tumors with this in mind. However, despite these limitations, our study has successfully classified patients that failed initial chemotherapy for HGSC based on clinical and molecular characteristics and postulated new targeted treatment strategies for patients with very poor outcomes that have few therapeutic options. 

## 4. Material and Methods

### 4.1. Outcomes Definition 

Patients were categorized as responders or non-responders. The responders were defined as those with progression-free survival six months after the completion of six cycles of platinum-based chemotherapy. Non-responders were those who had progressed during the first platinum-based chemotherapy (platinum-refractory) or those who recurred within six months of treatment completion (platinum-resistant) [34,35,36]. Data from 450 patients with serous epithelial ovarian, fallopian tube, or primary peritoneal cancer were extracted from TCGA. The clinical characteristics of the study cohort are shown in Table 1. There was a total of 292 responder patients and 158 non-responders.

### 4.2. Source of Data 

TCGA genomic data, including copy number variation, single nucleotide polymorphisms (SNPs), miRNA expression, gene expression (mRNA), and DNA methylation, as well as clinical and outcome information, were downloaded, normalized, formatted, and organized for the analysis, in accordance with the precepts of the TCGA data sharing agreements. All data collection and processing, including the consenting process, were performed after approval by the University of Iowa Institutional Review Board and they were in accord with the TCGA Human Subjects Protection and Data Access Policies, adopted by the National Cancer Institute (NCI) and the National Human Genome Research Institute (NHGRI).

#### 4.2.1. Copy Number Alterations (CNA) 

Samples from Agilent Human Genome CGH Microarray 244A (Agilent Technologies, Santa Clara, CA) were processed and DNA sequences were aligned to NCBI Build 36 version of the human genome. Circular Binary Segmentation was used to identify the regions with an altered copy number in each chromosome [37]. The copy number at a particular genomic location was computed based on the segmentation mean log ratio data. We found regions with frequent CNA among all the samples by performing genomic identification of significant targets in cancer (or GISTIC) analysis [38]. The significance of CNA at a particular genomic location was determined based on false discovery rate (FDR), as previously described [39]. A total of 16,918 chromosomal loci were included in the analysis. CNA was available for 447 patients (CR = 290, IR = 157).

#### 4.2.2. Mutation Analysis 

Somatic mutation detection, calling, annotation, and validation have been extensively described elsewhere [39]. Somatic mutation information resulting from Illumina Genome Analyzer DNA Sequencing GAIIx platform (Illumina Inc., San Diego, CA) was downloaded and formatted for analysis. Mutation information was downloaded from TCGA Data Portal (Level 3, i.e., validated somatic mutations). Somatic mutation information was available from 171 samples from patients with CR and 89 with IR. For those patients, there were 6716 unique genes that presented some types of validated somatic mutation. These included: frame shift insertions and deletions, in-frame insertions or deletions, missense, nonsense and nonstop mutations, silence, splice site, and translation start site mutations.

#### 4.2.3. Gene Expression 

Raw gene expression data were downloaded from the TCGA Data Portal (Level 1), extracted, loaded, and normalized and annotated with the National Center for Biotechnology Information (NCBI) Build 36 of the human genome. There were 450 (CR = 292, IR = 158) Affymetrix HT Human Genome U133 arrays with gene expression and clinical information about the chemo-response. A total of 12,718 genes passed the filtering criteria for percentage of values missing (<50%) and they were included in the analysis.

#### 4.2.4. DNA Methylation 

DNA methylation data with beta-values and methylated (M) and unmethylated (U) intensities were downloaded from the TCGA Data Portal (Level 2), extracted, loaded, and normalized. There were 438 (CR = 286, IR = 152) unique DNA-methylation of Illumina Infinium Human DNA Methylation 27 (Illumina Inc., San Diego, CA, USA) arrays from HGSC with clinical information regarding chemo-response. Differential DNA methylation of gene promoters was computed based on beta-values. The beta-values for each sample and locus were calculated as (M/(M+U)) [39]. A total of 14,473 DNA methylation probes passed filtering criteria for percentage of values missing (<50%) and they were included in the analysis.

#### 4.2.5. MicroRNA (miRNA) Expression 

Raw miRNA expression data were downloaded from the TCGA Data Portal (Level 1), extracted, loaded, and normalized with the analytical software, BRB-ArrayTools. There were 448 (CR = 290, IR158) Agilent Human miRNA Microarray Rel12.0 (Agilent Technologies Inc., Santa Clara, CA, USA) arrays from HGSC with clinical information regarding chemo-response [39]. A total of 619 miRNAs passed filtering criteria for the percentage of values missing (<50%) and they were included in the analysis.

### 4.3. Statistical Analysis 

#### 4.3.1. Variable Selection 

Our objective was to characterize HGSC patients that do not respond to initial chemotherapy by clustering clinical and biological data. Initially, we selected those variables that were associated with the defined outcome (non-response to chemotherapy) for all types of data: clinical, gene and miRNA expression, CNA, somatic mutations and DNA methylation. Variables that were not different between responders and non-responders do not inform the characterization of non-responder patients. 

For a selection of variables that are associated with non-response, we performed a univariate two-sided *t*-test analysis, comparing the groups of chemo-response (responders versus non-responders) with respect to differential gene expression, DNA methylation, miRNA expression (cut off for significance *p*-value < 0.05), and CNA (*p*-value < 0.001). Logistic regression analyses were performed to determine the association between chemo-response, somatic gene mutations, and clinicopathological variables (cut off value for significance *p* < 0.05). A more stringent cut-off *p*-value was chosen for CNA (with elevated number of variables and overlap between them) to reduce false positives and minimize features in the cluster analysis. For all biological comparisons, 10,000 random permutations were performed to determine the *p* values. Significant variables in univariate analyses were included in the integrative cluster analysis, or *iClusterPlus* [40]. 

#### 4.3.2. *iClusterPlus* Analysis 

Only using data from the non-responder group, an unsupervised cluster analysis was performed with the *iClusterPlus* framework within the R statistical software package. The goal of this method is to generate a classification of tumors (or clusters) by capturing patterns from diverse classes of genomic data. The tumors were only clustered from non-responders. Initially, data from all biological classes were introduced in the model to optimize the clustering analysis. During the optimization process, no miRNAs made a significant contribution to the clustering solution; so, it was concluded that miRNA expression was not relevant to the clustering process. The remaining four biological/genomic classes were introduced for the final clustering analysis: gene expression, CNA, somatic mutations, and DNA methylation. All variables that were found to be significant in the univariate analysis were used. A total of 88 patients with information for all classes of biological data were used for the final analysis. 

Before performing the final *iClusterPlus* analysis, we optimized or tuned the parameters to be applied. First, we determined the number of clusters (k) by repeatedly partitioning the samples into a learning and a test set. Subsequently, we evaluated the degree of agreement between the predicted and the observed cluster assignment [40]. For results visualization, we plotted the number of clusters vs. percent of explained variation. The optimal k is the point at which the curve of the percent explained variation levels off [15]. Afterwards, for each k, we used Bayesian information criteria to select the best sparse model with the optimal combination of penalty parameters, or lambdas (ʎ) [40]. We ran the final model with the combination of optimized number of clusters (k), penalty parameters (ʎ), and all classes of variables. Finally, we selected the top features (95th percentile) that were based on lasso coefficient estimates for the k-cluster solution.

Important clinical information, including demographic data, stage, surgical, and treatment information, along with *TP53* mutation status was added to the representation of each cluster. The *TP53* mutations were grouped into three categories based on predicted functional consequence: gain-of-function (GOF), loss-of-function (LOF), or wildtype (WT). The GOF mutations were those that have been shown to cause an oncogenic phenotype: P151S, Y163C, R175H, L194R, Y220C, R248Q, R248W, R273C, R273H, R273L, and R282W [41]; LOF mutations were those that resulted in the lack of protein expression and WT mutations were those that did not alter the amino acid sequence of *TP53*. The remaining mutations were single missense mutations, or “variants of unknown significance”, in which the functional effect of the mutation is currently not known [42]. Residual disease after surgery was recorded in centimeters. Patients with residual disease >1 cm in largest diameter after surgery had suboptimal surgery. Optimal treatment was considered to be the sum of optimal surgery with the administration of six cycles of platinum-based chemotherapy.

Because miRNA expression did not contribute to the clustering process, in the representation of results, we only added miRNAs that were independently associated with chemo-response in the multivariate analysis. This analysis was performed using additive logistic regression modeling with backward elimination. 

#### 4.3.3. Pathway Analysis

To further characterize the molecular characteristics of each resulting cluster, a pathway enrichment analysis was performed using KEGG and *clusterProfiler* [18,43]. All of the analyses were performed using R environment for statistical computing and graphics (www.r-project.org) [44].

## 5. Conclusions

The integration of genomic and clinical data is a variable-based approach to cluster non-responders into distinct categories. Subsequent pathway analysis of the components of these clusters may be used to identify the potential alternative therapeutic targets and strategies for each cluster. 

## Figures and Tables

**Figure 1 ijms-20-01175-f001:**
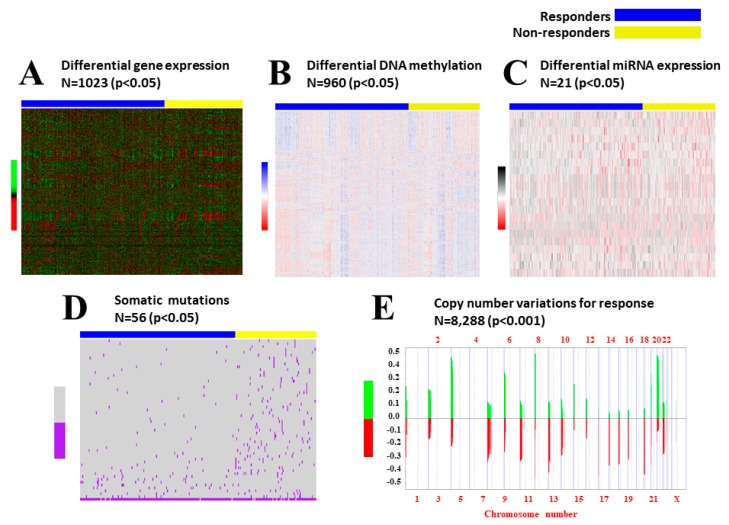
Univariate Analysis between Responders and Non-Responders. Heatmaps and graphics of molecular variables that were different between groups of responders and non-responders, with color codes and significance levels for each class of data. These variables were used for the integrative cluster analysis with *iClusterPlus*: (**A**) Differentially expressed genes; (**B**) Differentially methylated promoters; (**C**) Differentially expressed miRNAs; (**D**) Somatic mutations; (**E**) Altered gene copy numbers: green means gain of copy number, red is loss of copy number.

**Figure 2 ijms-20-01175-f002:**
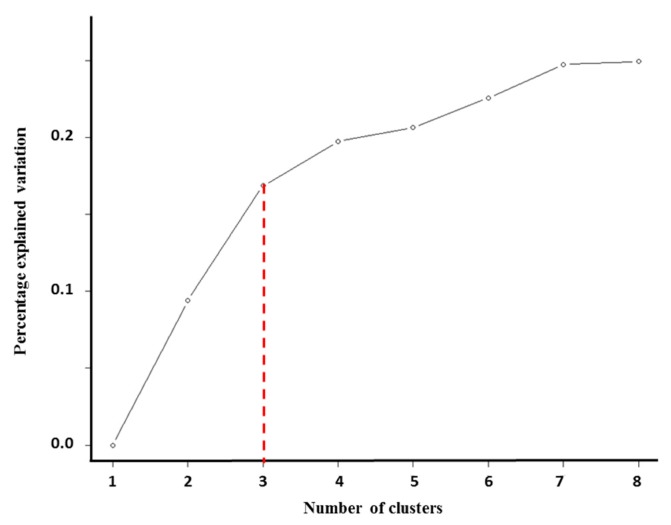
Optimization of cluster number. To assess the number of clusters we plotted the number of tested clusters vs. percent of explained variation. Optimal k or cluster number is the point at which percent of explained variation begins to level off after initial rapid ascent. Here, k = 3 [15].

**Figure 3 ijms-20-01175-f003:**
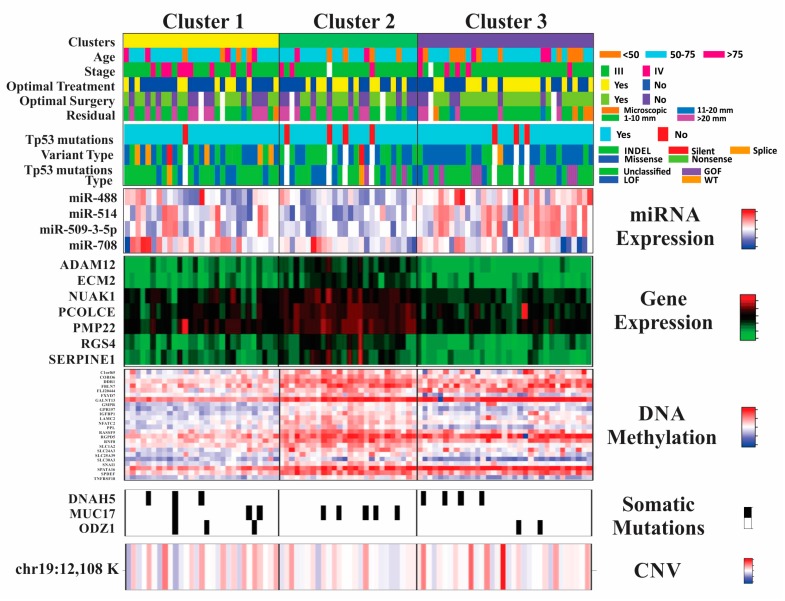
Clinical-molecular characteristics of the three clusters. At the top are the different clusters: 1 in yellow, 2 in green, 3 in purple. Below them are the clinical profiles with the variable in the left margin (age, stage, optimal treatment, optimal surgery and residual disease after surgery) and the color-code for each category in the right margin. Underneath clinical information there is a representation of *TP53* somatic mutation features: presence or status, variant, and mutation type. The last five heatmaps represent the top molecular features with specific color codes for their respective values at the right margin. Only molecular features that were most discriminating for this three-cluster model and passed a selection with a threshold value of > 95th percentile were included in the representation of the final 3-cluster model. Names of all features are detailed in Appendix A.

**Figure 4 ijms-20-01175-f004:**
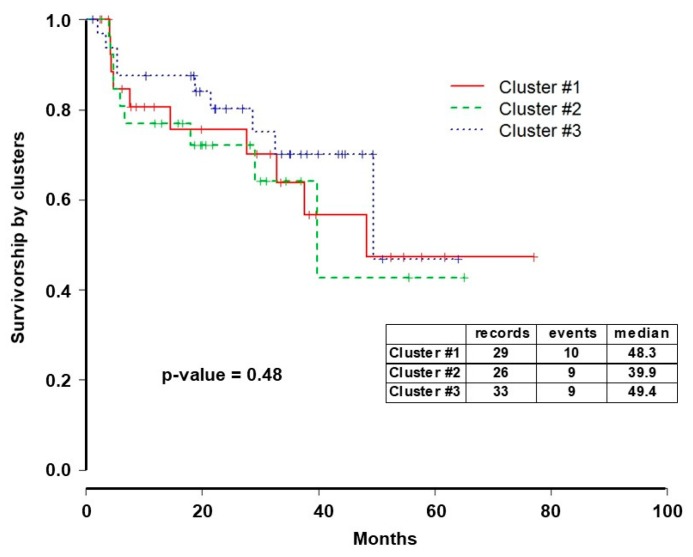
Survival analysis by clusters. Kaplan–Meier survival curves of the three clusters from *iClusterPlus* analysis showed no differences by log-rank analysis (*p* = 0.48).

**Table 1 ijms-20-01175-t001:** Clinical characteristics of 88 non-responder HSGC patients. Non-responders ^‡^ were divided by their resulting clusters after the *iClusterPlus* analysis. All of the clinical characteristics were not statistically different between the resulting clusters.

	Cluster #1	Cluster #2	Cluster #3	*p*-Value
Number of Patients	29	26	33	
Average Age (years)	61	59	57	0.149
Grade				0.744
Grade 2	2	4	3	
Grade 3	27	21	28	
Stage				0.081
Stage II	0	0	1	
Stage III	20	22	27	
Stage IV	9	3	5	
Surgical Outcome				0.079
Optimal (<1 cm)	15	13	24	
Suboptimal (>1 cm)	12	10	7	
Residual Disease				0.136
Microscopic	1	0	4	
Macroscopic	26	23	27	
Optimal Treatment				0.063
Optimal (Surgery + 6 cycles)	9	11	18	
Suboptimal	20	15	15	
Chemotherapy				0.151
Platinum	29	25 *	31 **	
Platinum +Taxane	27	24	31	

^‡^ Non-responders were those who had progressed during the first platinum-based chemotherapy (platinum-refractory) or those who recurred within 6 months of treatment completion (platinum-resistant). * One patient had no information about drugs delivered; all other had initial platinum-based chemotherapy. ** Two patients had no information about drugs delivered; all other had initial platinum-based chemotherapy.

**Table 2 ijms-20-01175-t002:** Pathway Enrichment Analysis. Given a list of genes, the pathway enrichment analysis using the Kyoto Encyclopedia of Genes and Genomes (KEGG) database will select those pathways that are overrepresented in the gene list for each one of the clusters. It will also compute a *p*-value for the resulting pathways. * Statistically non-significant.

KEGG ID	Description	*p*-Value	Gene ID
**Cluster 1**			
hsa04510	Focal Adhesion	<0.001	COL1A2/COL5A1/COMPATGA5/PDGFRA
hsa05214	Glioma	0.015	PDGFR1/PDGFRB/IGF1
hsa05218	Melanoma	0.019	PDGFR1/PDGFRB/IGF1
hsa05215	Prostate Cancer	0.034	PDGFR1/PDGFRB/IGF1
hsa04540	Gap Junction	0.034	PDGFR1/PDGFRB/PRKX
hsa05414	Dilated cardiomyopathy	0.034	ITGA5/PRKX/IGF1
hsa04512	ECM-Receptor Interaction	0.005	COL1A2/COL5A1/COMP/ITGA5
hsa04270	Vascular Smooth Muscle Contraction	0.013	ACTG2/CALD1/EDNRA/PRKX
**Cluster 2**			
hsa04218	Cellular senescence	0.001	NFATC2/RASSF5/SERPINE1/FBXW11
hsa04066	HIF-1 signaling pathway	0.040	SERPINE1/EGLN1
hsa00450	Seleno-compound metabolism	0.053 *	TXNRD2
hsa0 1040	Biosynthesis of unsaturated fatty acids	0.083 *	SCD5
hsa04390	Hippo signaling pathway	0.086 *	SERPINE1/FBXW11
hsa04310	Wnt signaling pathway	0.089 *	NFATC2/FBXW11
**Cluster 3**			
hsa04218	Cellular senescence	0.003	NFATC2/RASSF5/SERPINE1
hsa00512	Mucin type O-glycan biosynthesis	0.056 *	GALNT13

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
