# Peer review of "Molecular Characterization of Non-responders to Chemotherapy in Serous Ovarian Cancer"

_ijms, 2019, doi:10.3390/ijms20051175_

Reviewer 1 Report

This is a clinically relevant and timely article. Could the authors please consider the following:

-it would be useful to include an analysis of patients with optimal surgery/debulking, minimal residual disease, and optimal treatment since we already know that these factors when not optimal affect response to chemotherapy and survival. It is surprising that they did not find a significant difference in survival between the groups when these factors are taken into account.

-while the p values for the pathways highlighted in group 1 indicate significance, the p values for groups 2 and 3 are not less than 0.05. If this the case, they may not be good targets for the majority of patients

-could they please discuss more broadly the differences/pathways seen in each group? The scope of the work appears to be too narrow currently.

Author Response

IJMS- 443182: McDonald, et al., “Molecular Characterization of Non-responders to Chemotherapy in Serous Ovarian Cancer”

Response to Reviewers

We appreciate the reviewers’ careful reading and constructive critiques of our study to cluster ovarian cancer patients that did not respond to chemotherapy. The majority of comments were related to Table 1, which has now been modified as follows:

Limited the table features to the 88 non-responder patients;

Categorized patients by their respective cluster after iClusterPlus analysis;

Performed an association analysis between other clinical factors that may influence chemotherapy response and resulting clusters to assess whether there were any confounding variables;

Modified the table legend as recommended by Reviewer 3 and moved the highlighted text to the footnote of the Table.

 In addition, we have expanded the explanation of methods, results and the discussion of the various pathways identified in the three clusters. We believe these changes improve the clarity and interpretation of our data, and we feel we have completely addressed all reviewer critiques in this revised manuscript.   

Reviewer 1

This is a clinically relevant and timely article. Could the authors please consider the following:

1. It would be useful to include an analysis of patients with optimal surgery/debulking, minimal residual disease, and optimal treatment since we already know that these factors when not optimal affect response to chemotherapy and survival. It is surprising that they did not find a significant difference in survival between the groups when these factors are taken into account.

We appreciate this suggestion, which was also requested by other reviewers. As recommended, we have now modified the analysis to include an association analysis of the characteristics that also may influence response to chemotherapy, like optimal surgery, residual disease and optimal treatment. This analysis was conducted in the whole cohort of patients, including all types of responders to chemotherapy, and independently in non-responders to assess if there was any possible confounding factor that might bias the results. This new analysis is now detailed in Table 1 and the Results section (Section 2.1. Data preprocessing). Clinical variables were not statistically different between the resulting clusters. Also, we re-checked the survival analysis for the 3 clusters and confirmed that there were no differences.

This reviewer also suggests that we assess why patients with ovarian cancer do not respond to initial treatment. While we agree such an analysis is worthy of study, our goal herein was to identify different groups of resistant patients that may benefit from alternative treatments. In a separate study, we are in the process of interrogating with more detail some of these events in ovarian cancer: optimal surgery, respond to treatment. However, but the design and methods used will be significantly different and we feel it is beyond the scope of the present manuscript to include such an analysis.

2. While the p values for the pathways highlighted in group 1 indicate significance, the p values for groups 2 and 3 are not less than 0.05. If this the case, they may not be good targets for the majority of patients

Since we initially performed the pathway enrichment analysis there have been updates in the KEGG (Kyoto Encyclopedia of Genes and Genomes), with the most recent update in February 2019. There have also been updates released for in the software used for the analysis: clusterProfiler, in December of 2018, and R statistical package. So, as a first step to answer the reviewer comment, we re-performed the analysis using the same input genes and the new software and improved databases. The results for Clusters 1 and 2 are more complete, with new pathways. For Clusters 2 and 3, some of the pathways now reach statistical significance. We have updated Table 2 with the new results. Also, we updated the text in the Results section (Section 2.3. Pathway enrichment analysis) and added a reference for clusterProfiler (Yu, G.; Wang, L. G.; Han, Y.; He, Q. Y., clusterProfiler: an R package for comparing biological themes among gene clusters. Omics: a journal of integrative biology 2012, 16, (5), 284-7).

We have chosen to include non-significant pathways because, in an expanded dataset, these trends may become significant and are thus worthy of consideration and inclusion in the present study. 

3. Could they please discuss more broadly the differences/pathways seen in each group? The scope of the work appears to be too narrow currently.

We appreciate this suggestion, which will improve the interpretation of our findings. We have made significant modifications to the Discussion section to better explain the various signaling pathways that were overrepresented in each cluster. We hypothesized that the combination of cluster analysis, clinical and mutational characteristics, and pathway analyses may reveal some of the biology behind each cluster. Indeed, pathway analysis of molecular characteristics can be used to gain insight into common biological processes that contribute to disease pathophysiology for each cluster [16]. For example, Cluster 1 seems to overrepresent pathways involved in growth factor signaling (PDGFR and VEGF), which contributes to cell proliferation. Cluster 2 was dominated by pathways that regulate cellular survival, particularly in the setting of stress (HIF-1 and Wnt/β-catenin signaling). Finally, the significantly altered pathway in Cluster 3 was related to cellular senescence. Not only do these distinct alterations hint at the potential mechanisms of chemoresistance, but they also can be used to suggest alternative treatments to override the dominant signaling pathways.  

Reviewer 2 Report

This manuscript entitled ‘Molecular Characterization Non-responders to Chemotherapy in Serous Ovarian Cancer’ is interesting in terms of characterizing serous ovarian cancer patients who do not respond to chemotherapy by clustering clinical and biological data. However, there are many points to be improved before acceptance.

Because the subjects of this study are non-responders to chemotherapy, Table 1 should be limited to eighty-eight non-responders and categorize according to the type of cluster.

Although the authors described the characteristics of three clusters in the section of Results and showed them in Figure 3, statistical significance is unclear.

In discussion, results of clinical trial of nintedanib are described. However, the results of clinical trials of nintedanib for platinum-resistant ovarian cancer are more informative.

Author Response

IJMS- 443182: McDonald, et al., “Molecular Characterization of Non-responders to Chemotherapy in Serous Ovarian Cancer”

Response to Reviewers

We appreciate the reviewers’ careful reading and constructive critiques of our study to cluster ovarian cancer patients that did not respond to chemotherapy. The majority of comments were related to Table 1, which has now been modified as follows:

Limited the table features to the 88 non-responder patients;

Categorized patients by their respective cluster after iClusterPlus analysis;

Performed an association analysis between other clinical factors that may influence chemotherapy response and resulting clusters to assess whether there were any confounding variables;

Modified the table legend as recommended by Reviewer 3 and moved the highlighted text to the footnote of the Table.

 In addition, we have expanded the explanation of methods, results and the discussion of the various pathways identified in the three clusters. We believe these changes improve the clarity and interpretation of our data, and we feel we have completely addressed all reviewer critiques in this revised manuscript.   

Reviewer 2

This manuscript entitled ‘Molecular Characterization Non-responders to Chemotherapy in Serous Ovarian Cancer’ is interesting in terms of characterizing serous ovarian cancer patients who do not respond to chemotherapy by clustering clinical and biological data. However, there are many points to be improved before acceptance.

1. Because the subjects of this study are non-responders to chemotherapy, Table 1 should be limited to eighty-eight non-responders and categorize according to the type of cluster. 

We appreciate this suggestion, and we have edited Table 1 to include only the 88 non-responders and categorized the cases by their cluster. Also, as requested by Reviewer 1 we included in the analysis optimal surgery/debulking, minimal residual disease, and optimal treatment as they may influence the response to chemotherapy.

The results are summarized in Table 1 and detailed in the Results section.

2. Although the authors described the characteristics of three clusters in the section of Results and showed them in Figure 3, statistical significance is unclear.

We apologize for the lack of clarity. We selected a threshold value to select those features that will be represented in the final 3-cluster model. This threshold selected the most discriminant features for the 3-cluster solution of the model. The threshold for this particular 3-cluster solution was placed at the 95th percentile. To better explain the cut-off used for the features selection, we have now added the following sentence to the Results section (2.2. Integrative iClusterPlus analysis) and legend of Figure 3:

‘To build the final model, we used a threshold, or cut-off value, that selected the most discriminative features for the 3-cluster model solution. The threshold was the 95th percentile. Only those features that passed this threshold were included in the representation of the final 3-cluster model.’

3. In discussion, results of clinical trial of nintedanib are described. However, the results of clinical trials of nintedanib for platinum-resistant ovarian cancer are more informative.

We have identified only one phase II clinical trial that evaluated nintedanib for the treatment of platinum-resistant ovarian cancer: NCT01669798. Unfortunately, final results have not been reported yet. The preliminary results posted at ClinicalTrials.gov state that the percentage of patients who survive progression-free for at least 6 months after initiating study therapy is about 11.5%. We will add a sentence to the Discussion section and the reference to this clinical trial to emphasize these preliminary results (Davidson, B. A.; Squatrito, R.; Duska, L. R.; Havrilesky, L. J.; Schwager, N.; McCollum, M.; Arapovic, S.; Secord, A. A., Phase II evaluation of nintedanib in the treatment of bevacizumab-resistant persistent/recurrent ovarian, fallopian tube, or primary peritoneal carcinoma. Journal of Clinical Oncology 2016, 34, (15_suppl), TPS5601-TPS5601).

Reviewer 3 Report

Please in Table 1 move "Responders are those with progression-free 77 survival 6 months after the completion of 6 cycles of platinum-based chemotherapy. Non-responders 78 were those who had progressed during the first platinum-based chemotherapy (platinum-refractory) 79 or those who recurred within 6 months of treatment completion (platinum-resistant)."

in the Legend Table (footer).

Author Response

IJMS- 443182: McDonald, et al., “Molecular Characterization of Non-responders to Chemotherapy in Serous Ovarian Cancer”

Response to Reviewers

Reviewer 3

1. Please in Table 1 move "Responders are those with progression-free survival 6 months after the completion of 6 cycles of platinum-based chemotherapy. Non-responders were those who had progressed during the first platinum-based chemotherapy (platinum-refractory) or those who recurred within 6 months of treatment completion (platinum-resistant)." in the Legend Table (footer).

We appreciate this suggestion to improve the readability of the table, and we have made the suggested change in the revised Table 1.

Round  2

Reviewer 1 Report

Gene pathways for which p values are not significant should not be included as key for various clusters. There is no basis to conclude that larger databases will find such pathways significant.

Author Response

Reviewer 1

Gene pathways for which p values are not significant should not be included as key for various clusters. There is no basis to conclude that larger databases will find such pathways significant.

As suggested, non-significant pathways have been excluded as key for any cluster. In the Results section (2.3. Pathway enrichment analysis, lines 135-136) we clarified that the Wnt was not significant. Also, in the Discussion section (lines 167-186), non-significant pathways were eliminated from our overall interpretation of pathway analyses results.

Finally, we corrected some of the potential targeted treatments for Cluster 2 centering in the significant pathways:

- HIF-1 signaling pathway and its relation with the PI3K/AKT/mTOR pathway;

- The antagonism of Wnt signaling to cellular senescence pathway could be leverage for potential novel treatments (lines 190-195). Added reference: Adams, P. D.; Enders, G. H., Wnt-signaling and senescence: A tug of war in early neoplasia? Cancer biology & therapy 2008, 7, (11), 1706-11.

Reviewer 2 Report

Thanks for your answering my comments.

I read all the author's comments and the revised manuscripts. Those were well written and good contents. It is suitable paper for adoption.

Author Response

Reviewer 2

I read all the author's comments and the revised manuscripts. Those were well written and good contents. It is suitable paper for adoption.

Thank you